Microbiology
Spectrum

# *Helicobacter pylori luxS* mutants cause hyperinflammatory responses during chronic infection

Christina Yang,[1] Alessandra Rodriguez y Baena,[2] Bryce A. Manso,[2] Shuai Hu,[1] Raymondo Lopez-Magaña,[1] Mané Ohanyan,[1] Karen M. Ottemann[1]

**ABSTRACT** *Helicobacter pylori* infects roughly half the world's population, causing gastritis, peptic ulcers, and gastric cancer in a subset. These pathologies occur in response to a chronic inflammatory state, but it is not fully understood how *H. pylori* controls this process. We characterized the inflammatory response of *H. pylori* mutants that cannot produce the quorum sensing molecule autoinducer 2 (AI-2) by deleting the gene for the AI-2 synthase, *luxS*. Our work shows that *H. pylori luxS* mutants colonize the stomach normally but recruit high numbers of CD4$^+$ T cells to the stomach during chronic infection. This increase in the number of CD4$^+$ T cells correlated with elevated expression of *CXCL9*, a chemokine important for T cell recruitment. Together, our results suggest that *H. pylori* may utilize AI-2 signaling to modulate the inflammatory response during chronic infection.

**IMPORTANCE** Many bacteria signal to each other using quorum sensing signals. One type of signal is called autoinducer 2 (AI-2), which is produced and sensed by the LuxS enzyme found in many bacteria, including the gastric pathogen *Helicobacter pylori*. *H. pylori* establishes chronic infections that last for decades and lead to serious disease outcomes. How AI-2 signaling and LuxS contribute to chronic *H. pylori* infection has not been studied. In this work, we analyzed how loss of *H. pylori*-created AI-2, *via* mutation of *luxS*, affects *H. pylori* chronic infection. *luxS* mutants did not have significant colonization defects, similar to their reported phenotype during early infection, but they did have high stomach levels of effector and regulatory T cells and T-cell-recruiting chemokines. These results suggest that *H. pylori* LuxS may play more of a role in modulating the immune response versus colonization.

**KEYWORDS** inflammation, chronic infection, ulcer, gastric cancer, quorum sensing

Quorum sensing is a cell-to-cell signaling system used to coordinate bacterial activities in response to population size. This process synchronizes behaviors that are energetically costly and inefficient when only a small percentage of the population engages in them. Quorum sensing relies on population size detection, which occurs by detecting bacterially synthesized signaling molecules called autoinducers (AIs). The most widely produced AI is AI-2. AI-2 is made when the enzyme LuxS catalyzes the cleavage of S-ribosylhomocysteine to produce L-homocysteine and 4,5-dihydroxy-2,3-pentanedione (DPD) (Fig. 1) (1, 2). DPD spontaneously cyclizes to form the quorum sensing molecule AI-2.

*Helicobacter pylori* is one of the many bacteria that can produce AI-2 (3). *H. pylori* is a Gram-negative bacterium that establishes chronic infection in the human stomach. This bacterium is believed to be acquired during childhood and persists for the individual's lifetime if not treated with antibiotics (4). Infected individuals develop a range of diseases. About 85% of people develop asymptomatic gastritis, 15% develop gastric or

Address correspondence to Karen M. Ottemann, ottemann@ucsc.edu.

The authors declare no conflict of interest.

**FIG 1** Biosynthesis of AI-2 and downstream responses to AI-2 (A) The activated methyl cycle (AMC) is the metabolic pathway in which AI-2 is generated. AI-2 is a byproduct of the AMC. 4,5-Dihydroxyl-2,3-pentadione (DPD) spontaneously cyclizes to form AI-2. (B) The characterized downstream responses to AI-2 in *H. pylori*.

duodenal ulcers, and about 1% develop gastric cancer, although rates of these diseases vary depending on geographic regions (5). Indeed, gastric cancer has a poor prognosis and is the fourth leading cause of cancer-related deaths worldwide (6). Thus, understanding *H. pylori* infection properties is critical to understanding these important diseases.

*H. pylori* produces and responds to AI-2. *H. pylori* produces AI-2 as many bacteria do, using the LuxS enzyme (3). *H. pylori* detects AI-2 using periplasmic binding proteins, like other well-studied microbes (7). In *H. pylori*, these are called AibA and AibB (8) (Fig. 1). Downstream of detection, AI-2 leads to chemotaxis and transcriptional responses in other microbes. In *H. pylori*, AibA and AibB act in a chemotaxis pathway with the chemoreceptor TlpB, which responds to AI-2 in an AibA- or AibB-dependent manner (8). In this chemotaxis pathway, AI-2 acts as a chemorepellent, promoting *H. pylori* to swim away from AI-2 (9). This response to AI-2 is similar to one seen in *E. coli*, in which the LsrB periplasmic binding protein binds and interacts with the chemoreceptor Tsr, but instead of repulsion, it facilitates chemotaxis towards AI-2 (7). *H. pylori* also mounts a transcriptional response to AI-2 that includes several flagellar genes described below (3, 10), but the molecular basis for this response, however, is not known. Thus, the main AI-2 signaling outcomes in *H. pylori* include chemotaxis and transcription, with known components of two periplasmic binding proteins and a chemoreceptor.

As mentioned above, AI-2 affects *H. pylori* gene expression. AI-2 modestly affects the expression of flagellar-related genes and concomitantly, affects motility (10, 11). Without *luxS,* there was decreased expression of flagellar genes, including the gene for the FlhA regulator, and less motility (10, 11). Motility defects seem to be strain dependent as some strains did not display these defects (12). AI-2 also affects CagA expression and translocation (13) as well as expression of urease through downregulating the orphan response regulator HP1021 (14). The AI-2 transcriptional regulation pathway, however, is not known. These results suggest that AI-2 has some gene expression effects in *H. pylori*, but the signaling mechanisms are not known.

A hallmark of *H. pylori* infection is its ability to maintain chronic infection. One ability that promotes this outcome is *H. pylori*'s immunomodulatory capacity, in which the microbe manipulates the immune response. While many factors feed into this ability, key factors include its ability to prevent innate immune recognition (15), promote a T-regulatory state (16), and regulate bacterial population size. This latter response has been detailed in *H. pylori* mouse infections, in which the bacterium initially colonizes to high levels, but after about 2 months, bacterial numbers decrease and remain relatively low (17, 18). Thus, immune control and population maintenance are two key attributes that promote *H. pylori* chronicity.

Although AI-2 has been well documented to regulate bacterial responses, recent studies also suggest that AI-2 in several microbes can alter host immune responses. *E. coli* strains that produce varying amounts of AI-2 alter host immunity-related gene expression in cultured epithelial cells in a manner that correlated with AI-2 levels (19). These authors showed that AI-2 was responsible for these gene expression responses by using synthetic AI-2, supporting the hypothesis that these pathways responded to AI-2 directly

at the transcriptional level (19). Similarly, AI-2 from *Fusobacterium nucleatum* affected protein expression of macrophages (20). The most upregulated protein was TNFSF9, whose function is associated with the regulation of inflammatory responses, apoptosis, and tumorigenesis (20). These studies suggest that AI-2 can affect host gene expression, in addition to genes of bacteria, with documented effects on immune responses.

Whether and how AI-2 signaling and LuxS contribute to *H. pylori* infection have remained unclear. During acute single-strain infections, *H. pylori luxS* mutants did not have colonization defects in either mice or gerbils (12, 21). However, competition infections, using wild-type (WT) and *luxS* mutants, suggested that *luxS* contributes to colonization of some strains but not others (12). There have been no studies of how LuxS affects colonization and pathogenesis in a chronic infection model. We asked, therefore, how loss of *luxS* would affect *H. pylori* colonization and the host immune response in *H. pylori* chronic infections. Similar to findings in acute infections, *luxS* mutants did not have significant colonization defects. They did, however, lead to altered immune responses, specifically with higher numbers of effector and regulatory T cells in the stomach compared with WT *H. pylori* infections. Similarly, *luxS* mutants elicited mice to express high levels of chemokines involved in T-cell recruitment. These results suggest that *H. pylori* LuxS plays an important role in allowing *H. pylori* to modulate the immune responses.

## RESULTS

### *H. pylori luxS* mutants colonize to wild-type levels during late infections

To test the role of quorum sensing during chronic *H. pylori* infection, we generated a strain of *H. pylori* that does not produce AI-2 by deleting the gene encoding the LuxS AI-2 synthase and replacing it with a gene conferring erythromycin resistance. This mutant was created in the *H. pylori* SS1 background, a commonly used model strain that robustly and chronically infects mice (22). To confirm that the *H. pylori* SS1 strain *luxS* mutant did not produce AI-2, we used the *Vibrio harveyi* TL26 AI-2 bioluminescent reporter strain (23). This *V. harveyi* strain cannot produce AI-2; therefore, it only luminesces in the presence of exogenous AI-2. *H. pylori* SS1 WT and *luxS* strains were cultured overnight for 16 h, supernatants collected, and cells removed. Cell-free supernatant from WT *H. pylori* SS1 resulted in significantly higher luminescence compared with the media-only control (Fig. 2A). The isogenic *luxS* mutant, by comparison, produced background levels of luminescence that were similar to the media-only control. These data confirm that *H. pylori* SS1 produces AI-2 in a LuxS-dependent manner (Fig. 2A).

Next, mice were infected with either *H. pylori* SS1 WT or *luxS* mutants for 6 months. After 6 months, the stomachs of the mice were isolated and sectioned into the corpus and antrum regions. Each section was homogenized and plated for colony forming units. Both strains colonized robustly, with no significant difference in number of bacteria between the two infection groups in either region of the stomach (Fig. 2B). These data support that LuxS does not significantly affect *H. pylori*'s ability to colonize the stomach, similar to what was shown in acute infections (12, 21).

### *H. pylori luxS* mutants recruit more T cells to the stomach during chronic infection

We next analyzed how the host responded to *H. pylori* strains that lack *luxS*. T cells are an essential part of the chronic immune response the host generates against *H. pylori* (24). Therefore, we examined the stomach T-cell compartment by flow cytometry (Fig. S1). Mice infected with *luxS* mutants had higher numbers of αβ T cells in the stomach compared with mice infected with WT *H. pylori* (Fig. 3A). Among the αβ T cells, we also examined CD4$^+$ and CD8α$^+$ T cells. Mice infected with *luxS* mutants had significantly higher numbers of CD4$^+$ T cells (Fig. 3B) and trended towards having higher numbers of CD8α$^+$ T cells (Fig. 3C) in the stomach than those infected with WT *H. pylori*. We then examined the three CD4$^+$ T cell subsets, Th1, Th17, and Tregs, which are known

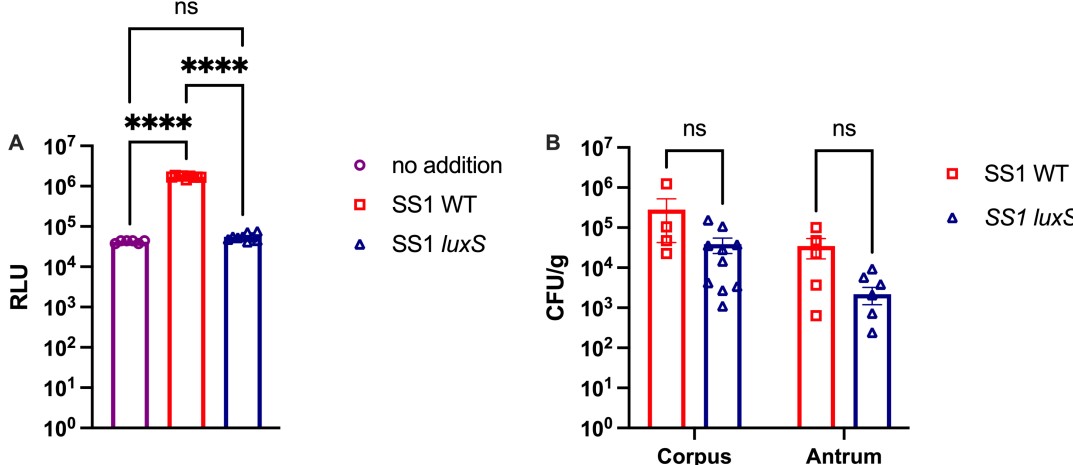

**FIG 2** *H. pylori* SS1 *luxS* mutants do not produce AI-2 and do not have colonization defects *in vivo*. (A) Relative light units (RLUs) indicative of AI-2 amounts in the cell-free culture supernatant *of H. pylori* SS1 WT or *luxS* grown in BB10 media overnight. AI-2 RLU was measured by exposing the AI-2 bioluminescent reporter *V. harveyi* TL26 to *H. pylori* cell-free media supernatant from WT or *luxS* mutant or a negative control of media alone. Each point represents technical replicates from a representative biological replicate of three biological replicates, and error bars indicate standard deviation. Statistical analyses were performed using one-way ANOVA with Tukey's multiple comparison test, indicated as *, $P < 0.05$; **, $P < 0.01$; ***, $P < 0.001$; ****, $P < 0.0001$. (B) Female C57Bl6/N mice were infected with *H. pylori* SS1 strains for 6 months. The total number of bacteria in the corpus and antrum of the stomach of infected mice was quantified by homogenizing tissue and plating for colony forming units (CFU) and normalized to the weight of the tissue. Each point represents tissue from one mouse, and error bars represent the standard error of the mean. Statistical analyses were performed using one-way ANOVA with Tukey's multiple comparison test, indicated as ns = not significant.

to be important in the immune response against *H. pylori* (25–27). Th1, Th17, and Treg cell compartments were examined by intracellular cytokine staining for IFNγ, IL17α, and IL-10, respectively. Overall, WT infected mice had a robust recruitment of pro-inflammatory Th1 (CD4+ IFNγ+ ) and Th17 (CD4+ IL17α) cells to the stomach, as compared to uninfected animals (Fig. 4). Compared with WT, *luxS*-infected mice had significantly higher numbers of Th1, Th17, and Treg (CD4+ IL10+) cells in the stomach (Fig. 4). Together, these data suggest that T cells are recruited to the stomach in response to both WT and *luxS* mutant *H. pylori*, but the *luxS* mutants have a more robust recruitment of T cells to the stomach in response to chronic infection.

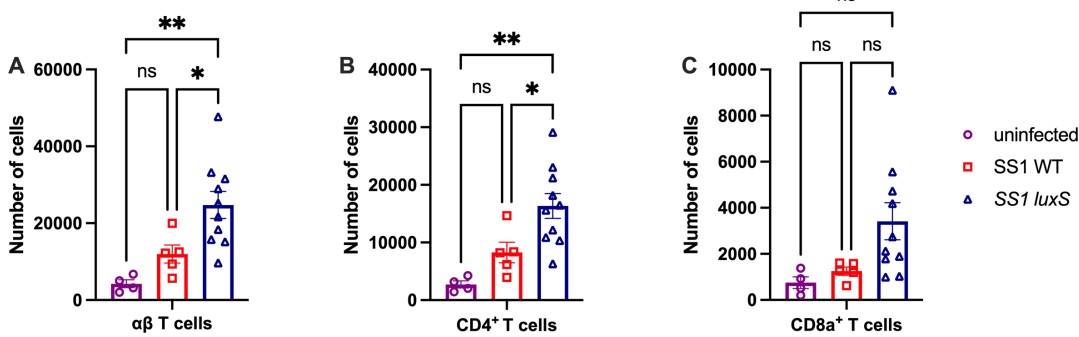

**FIG 3** *H. pylori luxS* mutants recruit more T cells to the stomach of infected mice compared with WT *H. pylori* female C57/BL6N mice were infected with SS1 WT or SS1 *luxS,* and lymphocytes from the corpus lamina propria were isolated at 6 months post-infection (n = 5–10). Mice same as shown in Fig. 1B. Analysis includes uninfected mice. (A) αβ T cells, (B) CD4+ T cells, and (C) CD8α+ T cells were analyzed by flow cytometry using the gating strategy shown in Fig. S1. Error bars represent the standard error of the mean. Statistical analyses were performed using one-way ANOVA with Tukey's multiple comparison test, indicated as *, $P < 0.05$; **, $P < 0.01$; ***, $P < 0.001$; ****, $P < 0.0001$.

## *H. pylori luxS* mutants affect gastric chemokine expression

T cells are primed in the lymph nodes and other lymphoid organs to pathogen-specific antigens and then are recruited to the site of infection by chemokine gradients (28). Because there were higher numbers of T cells in the stomach of mice infected with *luxS* mutants, we investigated whether there was a difference in the expression of chemokines involved in T-cell migration, specifically *Cxcl9* and *Cxcl10*. We did not look at the chemokine *Cxcl11* because it is not expressed in C57BL/6 mice (29). RNA was isolated from infected or uninfected mouse stomachs, and the abundance of chemokines *Cxcl9* and *Cxcl10* (30) was determined. Mice infected with WT *H. pylori* had significantly higher expression of *Cxcl9* and *Cxcl10* compared with mock-infected mice (Fig. 5). Consistent with *luxS*-infected mice having higher numbers of T cells in the stomach, these mice had higher expression of *Cxcl9*, but not *Cxcl10*, compared with WT-infected mice. These results suggest that the host develops a different chemokine profile in response to *luxS* mutants compared with WT.

## DISCUSSION

LuxS is a bacterial enzyme that produces the quorum sensing molecule AI-2. AI-2 is used by bacteria to sense bacterial population size and coordinate processes as a community rather than on a single-cell level. Here, we report that in a chronic model of infection, *luxS* mutants recruit higher numbers of T cells than WT *H. pylori*. These mutants did not have significant colonization differences 6 months post-infection (Fig. 2B). Despite no significant colonization differences, *luxS* mutants recruited significantly more CD4$^+$ T helper cells to the stomach than WT *H. pylori* (Fig. 4), which coincided with increased expression of *CXCL9*, a chemokine involved in T-cell recruitment (Fig. 5).

Our results are consistent with the idea that *H. pylori luxS* mutants cause more inflammation than WT. A similar finding has been observed in *Pseudomonas aeruginosa* (31). *lasR* mutants that cannot produce the quorum sensing molecule N-acyl homoserine lactone have a hyperinflammatory phenotype associated with increased cytokine expression and increased neutrophil recruitment (31). The authors found that these mutants fail to express LasB, a protease, that normally degrades the cytokine IL-8. This elevated IL-8, in turn, was thought to underlie the observed increased neutrophil recruitment (31). Therefore, it is possible that *H. pylori luxS* mutants cause more inflammation than WT *H. pylori* because AI-2 is used to regulate expression of *H. pylori* genes that would regulate the immune response. Of note, the mutant used here has not been complemented; however, *luxS* is predicted to be the last gene in its operon and has its own transcriptional start site, suggesting that the mutation will not have polar effects

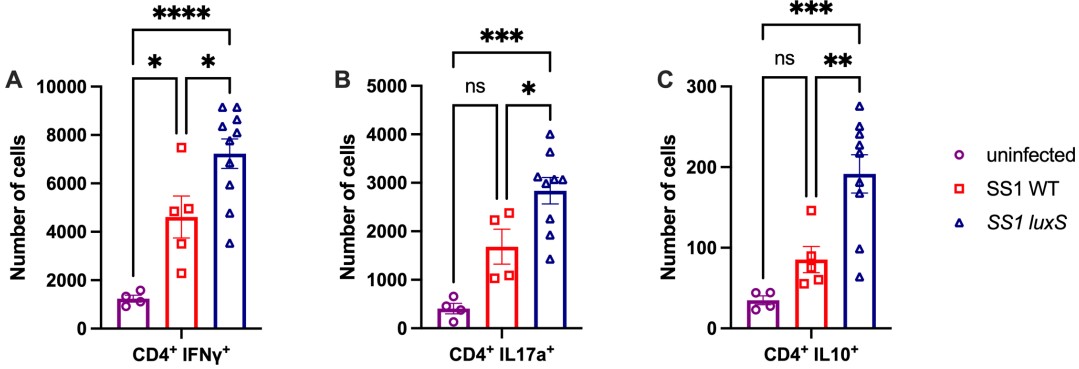

**FIG 4** *H. pylori luxS* mutants recruit more effector and regulatory T cells to the stomach compared with WT *H. pylori* female C57/BL6N mice infected with *H. pylori* SS1 WT or SS1 *luxS* and lymphocytes from the corpus lamina propria were isolated at 6 months post-infection (n = 5–10). Analysis includes uninfected mice. (A) CD4$^+$ IFNγ, (B) CD4$^+$ IL17α, and (C) CD4$^+$ IL10 were analyzed by flow cytometry as shown in Fig. S1. Mice same as shown in Fig. 1B. Error bars represent the standard error of the mean. Statistical analyses were performed using one-way ANOVA with Tukey's multiple comparison test, indicated as *, $P < 0.05$; **, $P < 0.01$; ***, $P < 0.001$; ****, $P < 0.0001$.

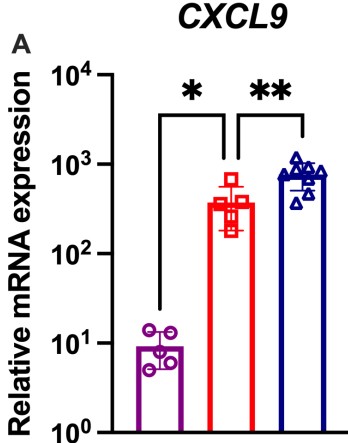

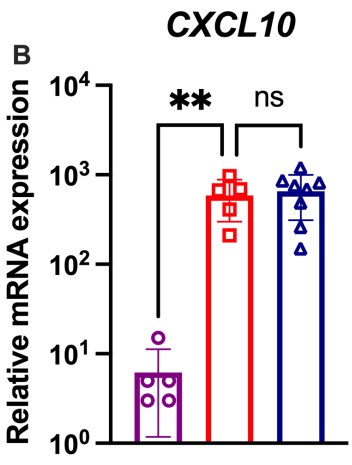

**FIG 5** *H. pylori luxS* alters the expression of immune signaling molecules *in vivo*. Gastric mRNA expression of (A) *Cxcl9* and (B) *Cxcl10* in infected mice was determined by isolating RNA from corpus tissue and using it for qRT-PCR. mRNA expression levels are presented as relative mRNA expression normalized to *Gapdh*. Mice same as shown in Fig. 1B. Error bars represent the standard error of the mean. Statistical analyses were performed using one-way ANOVA with Tukey's multiple comparison test, indicated as *, $P < 0.05$; **, $P < 0.01$; ***, $P < 0.001$; ****, $P < 0.0001$.

(32). *In vitro* experiments have shown that AI-2 in *H. pylori* regulates flagellar motility genes in some strains, with *luxS* mutants having decreased motility (10, 11). This phenotype would not be expected to cause increased T-cell recruitment, as other motility mutants (*e.g.*, those lacking chemotaxis) have decreased T cell numbers (33). AI-2 has been shown to affect expression of a known pro-inflammatory gene, *cagA* (13), but the strain used here does not have the ability to translocate CagA due to inactivation of the type 4 secretion system (34), so this effect would not be predicted to increase T cell numbers. Finally, AI-2 has been shown to downregulate HP1021, which encodes an orphan transcriptional response regulator (14). HP1021 was shown to alter the expression of multiple genes, including the *ureAB* genes, which encode for the virulence factor urease. One possible model for future testing is that WT *H. pylori* creates AI-2 that in turn acts *via* HP1020 to regulate genes that dampen inflammation; without AI-2, *luxS* mutants elevate inflammation because they lose regulation of key *H. pylori* genes.

Another possible explanation for why *luxS* mutants cause more inflammation than WT *H. pylori* is that AI-2 directly alters the host immune response. Zargar *et al.* showed that different strains of *E. coli* that produce varying amounts of AI-2 alter gene expression in the intestinal epithelial cell line HCT-8. Specifically, they observed that many pathways that were affected were related to the host immune response (19). Additionally, Wu *et al.* showed that AI-2 affected protein expression in macrophages. The most upregulated protein was TNFSF9, whose function is associated with the regulation of inflammatory responses, apoptosis, and tumorigenesis (20). Although limited, these studies suggest that AI-2 could directly affect the host immune response by affecting the expression of immune-related genes and proteins. This could offer one possible explanation for our results that *luxS* mutants do not produce AI-2, which could directly affect the expression of *Cxcl9* expression. CXCL9 alone could cause differences in the recruitment of T cells to the site of infection. CXCL9, CXCL10, and CXCL11 all bind to the same chemokine receptor CXCR3. However, they bind this receptor with different affinities and lead to different outcomes. CXCL9 binding to CXCR3 specifically promotes CD4[+] T cell polarization towards Th1/Th17 (30). This is consistent with our results that there are significantly more Th1 and Th17 cells than Treg cells in the *H. pylori ΔluxS*-infected stomach (Fig. 4).

In sum, our work agrees with previously published studies that show *H. pylori luxS* and AI-2 is not required for mouse colonization. *luxS* mutant strains, however, display elevated *Cxcl9* and concomitant increase in T cell numbers recruited to the gastric tissue. This finding suggests that *H. pylori* AI-2 normally acts either directly or indirectly to dampen the immune response during chronic infection.

## MATERIALS AND METHODS

### Bacterial strains and culture conditions

*H. pylori* strains used in this study were derived from the mouse derivative strain SS1. They include WT GFP$^+$ (17) and SS1 Δ*luxS* GFP$^+$ (this study) (Table 1).

*H. pylori* strains were grown at 37°C under microaerophilic conditions of 5% $O_2$ and 10% $CO_2$ on solid media consisting of Columbia blood agar (BD Diagnostics, Fisher Scientific) with 5% defibrinated horse blood (Hemostat Laboratories, Dixon, CA), 50 µg/mL cycloheximide (VWR), 10 µg/mL vancomycin, 5 µg/mL cefsulodin, 2.5 U/mL polymyxin B (all from Gold Biotechnology, St. Louis, MO), and 0.2% (wt/vol) β-cyclodextrin (Spectrum Labs, Gardena, CA) (CHBA) or in liquid medium consisting of Brucella Broth medium supplemented with 10% heat-inactivated fetal bovine serum (FBS) (Life Technologies) (BB10) under shaking conditions.

SS1 Δ*luxS* was generated by disrupting the gene with an erythromycin antibiotic resistance cassette. The deletion construct was generated by Gibson Assembly: the erythromycin resistance gene cassette from strain pErm-T-tnpR was flanked by 500 bp upstream and downstream of the *luxS* gene using primers listed in Table 1. This construct was designed to replace the entire protein coding sequence of the *luxS* gene. PCR amplicons were introduced into SS1 by natural transformation, and the *luxS* mutant in which the endogenous gene was disrupted by the insertion of the erythromycin resistance gene was selected using the CHBA supplemented with erythromycin (1 µg/mL) and verified by PCR genotyping. To generate fluorescent *H. pylori*, strains were transformed with pTM115 as done previously, to kanamycin resistance (17).

*V. harveyi* strain TL26 was generously provided by the Bassler lab (Princeton, New Jersey). TL26 was grown in Autoinducer Bioassay (AB) agar or broth (1.753% NaCl, 0.602%, $MgSO_4$, 0.2% casamino acids, 1%K 3 PO, 1% 0.1M L-arginine, 1% glycerol) (35) with shaking at 30°C under atmospheric conditions

### Luminescence measurements

An overnight culture of *V. harveyi* TL26 in AB media was diluted at 1:2,500 with AB broth, and 90 µL was aliquoted to white-walled, clear, flat bottom 96-well plates (Corning). *H. pylori* strains were grown for 16 h in BB10 with shaking, as above. Cell-free supernatant from desired overnight-grown *H. pylori* strains was prepared by filtration through a 0.22 µM filter (Millipore). Then, 10 µL of cell-free supernatant was added to each well and incubated at shaking conditions at 30°C. After 4 h, luminescence and $OD_{600}$ were measured using a plate reader (Perkin Elmer Victor X3).

### Animal infections

Female C57BL/6N mice (*Helicobacter* free, Charles River) at 6–8 weeks of age were orally infected by pipette feeding with *H. pylori*, as done previously (36, 37). *H. pylori* SS1

**TABLE 1** Bacterial strains

| KO | Strain | Description | Source | Primers used |
|---|---|---|---|---|
| 457 | SS1 | | (22) | |
| 529 | SS1 GFP$^+$ | pTM115 | (17) | |
| 1838 | SS1 *luxS* | Δ*luxS::erm* | This study | **CY58**: cgaggtcgacggtatcgataATAGATATTGTTTTGGTGTACAATAAG |
| | | | | **CY59**: tcattttcatCTCATTCTCCTATTCTTAAAGTG |
| | | | | **CY60**: ggagaatgagATGAAAATGAATGTAGAGAGTTTC |
| | | | | **CY61**: cttataaaatCATGAAATTAACGCTGATG |
| | | | | **CY62**: ttaatttcatgATTTTATAAGGAGGAAAAAATAAAGAG |
| | | | | **CY63**: gctgcaggaattcgatatcaAAACAAGTTAAGGATGCAG |
| 1831 | SS1 *luxS* GFP$^+$ | Δ*luxS::erm* pTM115 | This study | |
| | *V. harveyi* TL26 | ΔluxN, ΔluxS, ΔcqsS | (23) | |
| 1226 | *E. coli* pErm-T-tnpR | Plasmid containing erythromycin resistance gene | | |

**TABLE 2** Antibodies used for T-cell cytokine expression[a]

| Antibody | Anti | Fluorophore | Dilution |
| --- | --- | --- | --- |
| L/D fixable viability dye eF780 | - | APC-CY7 | 1:1000 |
| CD45 | Anti-mouse | PE | 1:200 |
| TCRβ | Anti-mouse | PE-CY7 | 1:200 |
| CD4 | Anti-mouse | APC | 1:200 |
| CD8α | Anti-mouse | PE-CY5 | 1:200 |
| CD3 | Anti-mouse | AF700 | 1:100 |
| IFNγ | Anti-mouse | BV421 | 1:50 |
| IL-10 | Anti-mouse | BV605 | 1:50 |
| IL-17α | Anti-mouse/rat | AF488 | 1:50 |

[a]"-" indicates L/D fixability dye eF780 is not an antibody but a dye, so there is no secondary antibody.

strains were prepared by growing to between mid-exponential phase ($OD_{600}$ = ~0.3) to late exponential phase (OD600 = ~0.75) in BB10 overnight and were checked for GFP fluorescence and motility using a Nikon Eclipse E600 phase-contrast microscope at 400× magnification with a LED illuminator (pE-300white, CoolLED) with a fluorescent filter for GFP. Motile GFP[+] cultures were concentrated to an $OD_{600}$ = 3 (~9 × $10^8$ CFU/mL) by centrifugation of 10 mL cultures in 15 mL Falcon tubes at 2,320×$g$ for 10 min. Culture supernatant was removed *via* aspiration, and pellets were gently resuspended in BB10 to concentrate the samples to an $OD_{600}$ = 3. For infection, mice were orally fed 50 µL of the $OD_{600}$ = 3 culture *via* a pipet tip for an inoculum of 4.5 × $10^7$ CFU. Input inoculum was plated on CHBA plates to confirm the input CFU. Between 7 and 21 mice were used for each strain. After the infection period, the mice were sacrificed by $CO_2$ narcosis. The stomach was removed at the stomach–esophageal junction and the antrum–duodenum sphincter, and then opened by cutting along the lesser curvature of the stomach. Stomach contents were removed gently by washing in PBS. The stomach was then separated between the antrum and corpus at the transition zone, based on tissue coloration. Each section was divided into pieces used to determine output CFU and for flow cytometry. To determine bacterial numbers, tissue pieces were weighed, homogenized using the Bullet Blender (Next Advance) with 1.0 mm zirconium silicate beads, and then plated onto CHBA plates, supplemented with 20 µg/mL bacitracin, 10 µg/mL nalidixic acid, and 15 µg/mL kanamycin to determine CFU/g of stomach tissue.

## Flow cytometry

Flow cytometry was carried out as described previously (38, 39). For cell surface staining, cells were stained in PBS with 0.5% BSA with the antibodies listed in Table 2. Mouse Fc Block (CD16/CD32, BD) was used to minimize non-specific antibody binding. Absolute cell counts were determined by adding a known number of CountBright Plus Absolute Counting Beads (Invitrogen) to each sample, which were analyzed on an LSRII (BD Biosciences). Up to 300,000 cells were recorded per mouse, with some samples containing as few as 30,000 cells. Analysis was performed using FlowJo (BD).

## RNA purification and quantitative PCR (qPCR)

Total RNA was extracted and purified from mouse gastric tissue, either non-infected or infected with different *H. pylori* stains using RNeasy Plus Mini Kit (QIAGEN) according to the manufacturer's instructions. cDNA was synthesized with 1 µg of RNA using

**TABLE 3** Primers for RT-qPCR

| Gene | Forward | Reverse |
| --- | --- | --- |
| CXCL9 | GAGCAGTGTGGAGTTCGAGG | TCCGGATCTAGGCAGGTTTG |
| CXCL10 | AATGAGGGCCATAGGGAAGC | AGCCATCCACTGGGTAAAGG |
| GAPDH | TGGAGAAACCTGCCAAGTATGA | CTGTTGAAGTCGCAGGAGACAA |

high-capacity cDNA reverse transcription kit (Applied Biosystems). RT-qPCR was then performed in triplicate using cDNA and SYBR Green PCR Master Mix (Applied Biosystems) in a CFX Connect Real-Time PCR Detection System (Bio-Rad). All primers for qPCR are listed in Table 3.

## ACKNOWLEDGMENTS

We thank Isabelle Arnold from the University of Zürich for her expertise and training on gastric sample flow cytometry; the laboratory of Prof. Bonnie Bassler (Princeton Univ.) for providing us with the *V. harveyi* strain TL26 to measure luminescence; Bari Holm Nazario and the UCSC Institute for the Biology of Stem Cells Flow Cytometry Facility for technical support and training (RRID: SCR_021149).

The described project was supported by the National Institute of Allergy and Infectious Diseases (NIAID) grant RO1AI116946 to K.M.O. A.R.y.B. is supported by NIH IRACDA 5K12GM139185-04 grant. B.A.M. is supported by an NIGMS K12GM139185 grant. The funders had no role in study design, data collection, and interpretation, or decision to submit the work for publication.

## AUTHOR AFFILIATIONS

[1]Department of Microbiology and Environmental Toxicology, University of California Santa Cruz, Santa Cruz, California, USA
[2]Department of Biomolecular Engineering, University of California Santa Cruz, Santa Cruz, California, USA

## AUTHOR ORCIDs

Christina Yang  http://orcid.org/0000-0002-5917-9338
Karen M. Ottemann  http://orcid.org/0000-0001-6265-7401

## AUTHOR CONTRIBUTIONS

Christina Yang, Conceptualization, Formal analysis, Investigation, Methodology, Writing – original draft, Writing – review and editing | Alessandra Rodriguez y Baena, Investigation, Writing – review and editing | Bryce A. Manso, Investigation, Writing – review and editing | Shuai Hu, Investigation, Writing – review and editing | Raymondo Lopez-Magaña, Investigation, Writing – review and editing | Mané Ohanyan, Investigation, Writing – review and editing.

## ETHICS APPROVAL

The University of California Santa Cruz Institutional Animal Care and Use Committee approved all animal protocols and experiments performed (Protocol Ottek 2014). Female C57BL/6N mice (*Helicobacter* free, Charles River) were housed and cared for at the University of California, Santa Cruz Vivarium.

## ADDITIONAL FILES

The following material is available online.

### Supplemental Material

**Figure S1 (Spectrum01073-24-s0001.pdf).** Gating strategy.

### Open Peer Review

**PEER REVIEW HISTORY (review-history.pdf).** An accounting of the reviewer comments and feedback.

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
