## [Reviewer comments · Microbiology Spectrum]

Microbiology Spectrum

***H. pylori luxS* mutants cause hyperinflammatory responses during chronic infection**

Christina Yang, Alessandra Rodriguez y Baena, Bryce Manso, Shuai Hu, Raymond Lopez Magaña, Mane Ohanyan, and Karen Ottemann

Corresponding Author(s): Karen Ottemann, University of California at Santa Cruz Department of Microbiology and Environmental Toxicology

Review Timeline:

Submission Date:	April 28, 2024
Editorial Decision:	June 18, 2024
Revision Received:	August 9, 2024
Accepted:	August 30, 2024

Editor: Jennifer Gaddy

Reviewer(s): The reviewers have opted to remain anonymous.

Transaction Report:

DOI: <https://doi.org/10.1128/spectrum.01073-24>

Re: Spectrum01073-24 (*H. pylori luxS* mutants cause hyperinflammatory responses during chronic infection)

Dear Prof. Karen M. Ottemann:

Thank you for the privilege of reviewing your work. Below you will find my comments, instructions from the Spectrum editorial office, and the reviewer comments.

Both Reviewers were enthusiastic about your manuscript and requested mostly clarification and text changes. Although one Reviewer did request complementation assays, and I agree this would be helpful, I do not consider it feasible with chronic infection models utilized in this manuscript and the resubmission timelines. I look forward to seeing your revised manuscript with the incorporated text and figure changes requested by both authors.

Revision Guidelines

Sincerely,
Jennifer Gaddy
Editor
Microbiology Spectrum

Reviewer #1 (Comments for the Author):

In this manuscript, Yang et al examine the impact of deletion of luxS, encoding AI-2 synthase, on *Helicobacter pylori* host colonization and immune response. In particular, they report increased recruitment of T cells, with a slight increase in the

expression of Cxcl9 in the gastric tissue upon infection with a H. pylori luxS deletion mutant versus wild type. I have one main question regarding the experiments:

Major comment:

- In the absence of complementation of the mutant, it is difficult to draw robust conclusions from the experiments presented. While the long-term nature of the infection experiment is acknowledged, inclusion of data with a complemented mutant strain for at least some of the key experiments is important.

Minor comments:

- It would be helpful for readers if some discussion regarding the increase in Cxcl9, but not Cxcl10, is added.
- Line 104: Missing the word "of" after "To test the role...".
- Reference(s) needed for the statements in lines 159-163.

Reviewer #2 (Comments for the Author):

Summary

In this article by Yang et al. entitled, "H. pylori luxS mutants cause hyperinflammatory responses during chronic infection" the authors present data that infection with a luxS mutant leads to increased effector T-cell recruitment to the stomach during a chronic infection mouse model with Helicobacter pylori. Mutation of luxS results in a lack of production of the quorum sensing molecule AI-2, thought to play a regulatory role in the inflammatory response. The authors first validate the luxS mutant does not produce AI-2 and has a similar colonization level as WT H. pylori during chronic infection. The authors then analyze isolated stomach tissues to reveal colonization with the luxS mutant leads to larger CD4+ T- cell populations and increased transcription of CXCL9, a chemokine involved in T-cell recruitment. The article is well written, and the use of a chronic infection model is particularly important for understanding long term implications of quorum sensing and immune regulation in H. pylori infection. The reviewer has a few comments and questions that might strengthen the quality of the paper.

Major

1. Can there be a general figure/graphic added to the introduction, visualizing some of the content from Lines 35-38 and 48-70? Specifically, including a general graphic for the biosynthesis of and downstream response to AI-2.
2. Was the transcriptional profile of CXCL11 investigated? If not, do the authors have a hypothesis as to how this chemokine might be impacted by luxS mutation? Given the different signaling cascade for this chemokine as opposed to CXCL9/10 this might be an interesting consideration to provide further insight into the immunomodulatory role of AI-2.
3. In the discussion it is slightly unclear how the authors propose AI-2 might be regulating the immune response to infection. The authors cite several roles of AI-2 that they do not think are responsible for modulating the observed inflammation in this study (e.g. citing opposite T-cell trends for other motility mutants, and lack of CagA translocation for this strain; Lines 165-172). However, the discussion is less clear about possible mechanisms that could be involved. It might be helpful to rewrite this section for better clarity on possible roles of AI-2 that are pertinent to this strain and the regulation of the inflammatory response investigated in this study.

Minor

1. In the figures, consider changing the yellow color of the SS1 luxS bar graphs to a color that is more visible/contrasting.
2. In Fig. 4A, is the statistical significance annotated correctly for uninfected vs SS1 WT?
3. A citation is needed for Line 159 for the statement "A similar finding has been observed in P. aeruginosa" and the discussion in the following sentences.
4. There are several typographical errors in the bibliography (e.g. Line 346-347, repeated DOI without journal name listed; Line 256, Journal name all capitalized; Line 343-345, no journal name listed). Please review the bibliography thoroughly to correct and make all references complete and uniform.

Editors comments

Thank you for the privilege of reviewing your work. Below you will find my comments, instructions from the Spectrum editorial office, and the reviewer comments.

Both Reviewers were enthusiastic about your manuscript and requested mostly clarification and text changes. Although one Reviewer did request complementation assays, and I agree this would be helpful, I do not consider it feasible with chronic infection models utilized in this manuscript and the resubmission timelines. I look forward to seeing your revised manuscript with the incorporated text and figure changes requested by both authors.

RESPONSE: Thanks for the top-quality reviews and understanding. We've addressed all the reviewers comments, including adding a brief model and explanation about the complementation. The manuscript is much strengthened, thanks!

Reviewer #1 (Comments for the Author):

In this manuscript, Yang et al examine the impact of deletion of *luxS*, encoding AI-2 synthase, on *Helicobacter pylori* host colonization and immune response. In particular, they report increased recruitment of T cells, with a slight increase in the expression of *Cxcl9* in the gastric tissue upon infection with a *H. pylori luxS* deletion mutant versus wild type. I have one main question regarding the experiments:

Major comment:

- In the absence of complementation of the mutant, it is difficult to draw robust conclusions from the experiments presented. While the long-term nature of the infection experiment is acknowledged, inclusion of data with a complemented mutant strain for at least some of the key experiments is important.

RESPONSE: This is a valid point, and we've now made the caveat clear in the Discussion at lines 159 as "Of note, the mutant used here has not been complemented, however *luxS* is predicted to be the last gene in its operon and has its own transcriptional start site, suggesting the mutation will not have polar effects (32)".

Minor comments:

- It would be helpful for readers if some discussion regarding the increase in *Cxcl9*, but not *Cxcl10*, is added.

RESPONSE: We have added discussion of this outcome in the Discussion, starting at line 179: "Although limited, these studies suggest that AI-2 could directly affect the host immune response by affecting the expression of immune-related genes and proteins. This could offer one possible explanation for our results that *luxS* mutants do not produce AI-2 which could directly affect the expression of *Cxcl9* expression. CXCL9 alone could cause differences in the

recruitment of T cells to the site of infection. CXCL9, CXCL10, and CXCL11 all bind to the same chemokine receptor CXCR3. However, they bind this receptor with different affinities and lead to different outcomes. CXCL9 binding to CXCR3 specifically promotes CD4+ T cell polarization towards Th1/Th17 30. This is consistent with our results that there are significantly more Th1 and Th17 cells than Treg cells in the *H. pylori* Δ luxS-infected stomach (Fig. 4).”

- Line 104: Missing the word "of" after "To test the role...".

RESPONSE: The word “of” has been added.

- Reference(s) needed for the statements in lines 159-163.

RESPONSE: Thank you for catching this, we have added the appropriate reference for these statements (section at current line 152).

Reviewer #2 (Comments for the Author):

Summary

In this article by Yang et al. entitled, "H. pylori luxS mutants cause hyperinflammatory responses during chronic infection" the authors present data that infection with a luxS mutant leads to increased effector T-cell recruitment to the stomach during a chronic infection mouse model with *Helicobacter pylori*. Mutation of luxS results in a lack of production of the quorum sensing molecule AI-2, thought to play a regulatory role in the inflammatory response. The authors first validate the luxS mutant does not produce AI-2 and has a similar colonization level as WT *H. pylori* during chronic infection. The authors then analyze isolated stomach tissues to reveal colonization with the luxS mutant leads to larger CD4+ T- cell populations and increased transcription of CXCL9, a chemokine involved in T-cell recruitment. The article is well written, and the use of a chronic infection model is particularly important for understanding long term implications of quorum sensing and immune regulation in *H. pylori* infection. The reviewer has a few comments and questions that might strengthen the quality of the paper.

RESPONSE: Thank you for the positive comments!

Major

1. Can there be a general figure/graphic added to the introduction, visualizing some of the content from Lines 35-38 and 48-70? Specifically, including a general graphic for the biosynthesis of and downstream response to AI-2.

RESPONSE: Good idea, we have added a new Figure to cover the AI-2 biosynthesis and downstream response as new Fig. 1, and then renumbered all the other Figures +1.

2. Was the transcriptional profile of CXCL11 investigated? If not, do the authors have a hypothesis as to how this chemokine might be impacted by luxS mutation? Given the different signaling cascade for this chemokine as opposed to CXCL9/10 this might be an interesting consideration to provide further insight into the immunomodulatory role of AI-2.

RESPONSE: No, the mouse strain we used does not produce CXCL11 and we have added that to the manuscript at line 136, as “We did not look at the chemokine Cxcl11 because it is not expressed in C57BL/6 mice (29)”.

3. In the discussion it is slightly unclear how the authors propose AI-2 might be regulating the immune response to infection. The authors cite several roles of AI-2 that they do not think are responsible for modulating the observed inflammation in this study (e.g. citing opposite T-cell trends for other motility mutants, and lack of CagA translocation for this strain; Lines 165-172). However, the discussion is less clear about possible mechanisms that could be involved. It might be helpful to rewrite this section for better clarity on possible roles of AI-2 that are pertinent to this strain and the regulation of the inflammatory response investigated in this study.

RESPONSE: Thank you for pointing this out. We have now added some ideas for what we think is happening in light of current knowledge at lines 168, as “One possible model, for future testing, is that WT *H. pylori* creates AI-2 that in turn acts via HP1020 to regulates genes that dampen inflammation; without AI-2, luxS mutants elevate inflammation because they lose regulation of key *H. pylori* genes”.

Minor

1. In the figures, consider changing the yellow color of the SS1 luxS bar graphs to a color that is more visible/contrasting.

RESPONSE: Thank you for this suggestion, we have modified the color to a blue in all figures.

2. In Fig. 4A, is the statistical significance annotated correctly for uninfected vs SS1 WT?

RESPONSE: We double checked this and yes, is annotated correctly (in current Fig. 5A).

3. A citation is needed for Line 159 for the statement "A similar finding has been observed in *P. aeruginosa*" and the discussion in the following sentences.

RESPONSE: Thank you for pointing this out, we've corrected this section as described for one of Reviewer 1's points.

4. There are several typographical errors in the bibliography (e.g. Line 346-347, repeated DOI without journal name listed; Line 256, Journal name all capitalized; Line 343-345, no journal name listed). Please review the bibliography thoroughly to correct and make all references complete and uniform.

RESPONSE: Thank you for pointing this out, we have carefully proofread the bibliography

Re: Spectrum01073-24R1 (*H. pylori luxS* mutants cause hyperinflammatory responses during chronic infection)

Dear Prof. Karen M. Ottemann:

Your manuscript has been accepted, and I am forwarding it to the ASM production staff for publication. Your paper will first be checked to make sure all elements meet the technical requirements. ASM staff will contact you if anything needs to be revised before copyediting and production can begin. Otherwise, you will be notified when your proofs are ready to be viewed.

Sincerely,
Jennifer Gaddy
Editor
Microbiology Spectrum